# Unique Four-Layer Core–Shell NaYF_4_:Yb^3+^,Er^3+^@NaYF_4_@CdS@Au Nanocomposites for Enhanced Full-Spectrum Photocatalytic Degradation of Rhodamine B

**DOI:** 10.3390/molecules30214215

**Published:** 2025-10-28

**Authors:** Yukun Tang, Pingping Yang, Jinpu Xie, Tengfei Duan, Zengmin Tang, Yao Liu, Rui Zhang, Haihu Tan, Jingjing Du, Lijian Xu

**Affiliations:** 1School of Packaging Engineering, Hunan University of Technology, Zhuzhou 412007, China; yukuntang0526@163.com (Y.T.); 18856765173@163.com (J.X.); tanhaihu2020@hut.edu.cn (H.T.); 2School of Biomedical Engineering, Hunan University of Technology, Zhuzhou 412007, China; yangpingping202211@163.com (P.Y.); duantengfei0809@163.com (T.D.); 14315@hut.edu.cn (Z.T.); zly04016@163.com (Y.L.); 15673070539@163.com (R.Z.)

**Keywords:** photocatalysis, CdS, Au nanoparticles, RhB degradation, four-layer core–shell structure

## Abstract

In recent years, cadmium sulfide (CdS) has been widely investigated due to its excellent photocatalytic performance. However, its practical application in pollutant treatment is limited by its narrow photoresponse range and susceptibility to photocorrosion. Herein, we design a unique four-layer core–shell structure NaYF_4_:Yb^3+^,Er^3+^@NaYF_4_@CdS@Au (CSNPs@CdS@Au), with an inert NaYF_4_ shell coating on NaYF_4_:Yb^3+^,Er^3+^ (CNPs) to form NaYF_4_:Yb^3+^,Er^3+^@NaYF_4_ (CSNPs) and CdS depositing on CSNPs (CSNPs@CdS); Au nanoparticles are loaded on CdS (CSNPs@CdS@Au). Compared with CdS (9.81%), CSNPs (5.0%), CSNPs/CdS (6.9%), and CSNPs@CdS (81.0%), CSNPs@CdS@Au degrades 97.7% Rhodamine B (RhB) within 15 min, exhibiting superior photocatalytic performance, attributable to two key factors: (1) the NaYF_4_ inert shell encapsulation amplifies upconversion (UC) luminescence intensity by suppressing surface quenching; and (2) the electron transfer between Au nanoparticles and CdS effectively promotes spatial separation of photogenerated charge carriers and increases reactive active sites. Additionally, after five degradation cycles, CSNPs@CdS@Au still maintains a 93.25% degradation rate for RhB, confirming its excellent stability. This remarkable stability is attributed to the uniquely designed multilayer core–shell architecture, which significantly enhances structural integrity through physical isolation effects. This study establishes a material preparation strategy for efficient photocatalytic pollutant degradation.

## 1. Introduction

Environmental pollution has recently emerged as a pressing global challenge, as the survival of humankind and societal development hinge upon the availability of clean freshwater [1,2,3]. Rhodamine B (RhB), a widely used organic dye, is one of the common sources of water pollution [4,5,6,7]. Solar-driven semiconductor photocatalytic technology can effectively harness light to degrade pollutants and is considered an efficient and eco-friendly water purification method [8,9]. Cadmium sulfide (CdS), as a type of semiconductor, has a narrow bandgap of 2.42 eV [10] and excellent optical performance [11], which demonstrates outstanding performance in the fields of degrading organic pollutants [12] and photocatalytic hydrogen production [13,14], and so on. However, traditional CdS photocatalysts can only absorb light with wavelengths below 520 nm and cannot effectively utilize near-infrared (NIR) light [15], which accounts for a relatively large proportion in the solar spectrum. Therefore, achieving the efficient utilization of CdS for NIR light has become a pressing problem.

At present, combining upconversion (UC) materials is regarded as an effective strategy to expand the absorption of the solar spectrum for CdS. Rare-earth (RE) ions in UC nanoparticles (UCNPs) possess rich f-electron configurations. Under NIR light, they leverage transitions between 4f and 4f-5d energy levels to convert low-energy NIR light into high-energy ultraviolet (UV) or visible light [16], thereby expanding the light absorption range of semiconductor materials. It is reported that the Yb^3+^-Er^3+^ ion pair doped in the β-NaYF_4_ has the highest UC efficiency [17]. Balaji et al. synthesized NaYF_4_:Yb/Er UCNPs for NIR light-driven photocatalytic degradation of methyl orange dye [18]. Feng et al. fabricated a NaYF_4_:Yb^3+^,Er^3+^/Au/CdS UC photocatalyst for photocatalytic hydrogen production from ethanol [19]. However, the energy transfer loss in UCNPs and surface quenching lead to a decline in UC efficiency, thus preventing the full expansion of the absorption range of photocatalysts. The introduction of an inert shell or energy-capturing ions to construct a core–shell structure is an effective way to suppress energy migration and surface quenching. Constructing an inert shell (NaYF_4_) on the surface to form a core–shell structure isolates the luminescent ions from solvents and ligands, thereby reducing the occurrence of fluorescence quenching. Rinkel et al. proposed a method for synthesizing 10 nm β-NaYF_4_:Yb,Er/NaYF_4_ core–shell UC nanocrystals. A shell approximately 2.5 nm thick enhanced the UC efficiency by approximately 160-fold [20]. Huang et al. coated a layer of NaYF_4_ shell on the UCNPs, resulting in a significant enhancement of the fluorescence effect [21]. In addition, the particle size of UC materials is also an important factor affecting their fluorescence effect [22,23]. It is reported that the fluorescence intensity of micrometer-sized UC materials is higher than that of nanometer-sized UC materials [24,25]. Li et al. synthesized core–shell structured NaYF_4_:Yb,Tm@CdS microrods (~6 μm in length). Due to the fact that the surface has not been functionalized, the adhesion effect of CdS on the NaYF_4_ surface is not good [22]. Li et al. synthesized NaYF_4_:Yb,Tm@CdS microrods of length ~1.5 μm. TEM characterization revealed that CdS were not uniformly distributed on the surface of the microrods [26]. On the one hand, CdS is prone to fall off from the UC rod-shaped material, resulting in unstable catalytic performance [13]. On the other hand, the uniform distribution of CdS on rod-shaped UC materials is difficult to control. Consequently, effectively integrating micrometer-scale UC rods with CdS remains challenging. In contrast, nanoscale UCNPs, particularly after coating with an inert shell, are more suitable for integration with CdS photocatalysts.

Furthermore, the practical application of CdS remains severely limited due to photocorrosion issues [27]. Photogenerated holes (h^+^) in the valence band (VB) of CdS oxidize S^2−^ within the material, following the reaction: CdS + 2h^+^→Cd^2 +^ +S [28,29]. Theoretically, multilayered core–shell structures can enhance structural stability through physical isolation. Combining CdS with noble metals (e.g., Au) to form core–shell architectures is considered one of the most effective methods for improving catalyst stability [30,31,32]. It has been reported that Li et al. constructed a ternary CdS@Au/MXene composite. The incorporation of Au established dual Schottky barriers, resulting in a catalytic rate 26.6 times higher than that of pure CdS [33]. Dong et al. prepared a four-layer core–shell structure of Co_3_O_4_@TiO_2_@CdS@Au double-layer nanocages, which achieved a removal rate of 98.8% for pollutants within 10 min. After five circles, it still demonstrated excellent catalytic performance and stability. More importantly, the photocorrosion of CdS was significantly suppressed, which is a key drawback of CdS-based photocatalysts [34]. Therefore, loading Au onto the surface of CdS to form a core–shell structure is expected to simultaneously enhance catalytic performance and stability [27,35].

Inspired by the above studies, this research aims to design a novel photocatalyst capable of efficiently utilizing the solar spectrum, suppressing CdS photocorrosion, and enhancing reusability. Our design involves the following steps: (1) Coating NaYF_4_:Yb^3+^,Er^3+^ (CNPs) with an inert NaYF_4_ shell to prepare NaYF_4_:Yb^3+^,Er^3+^@NaYF_4_ (CSNPs), converting NIR/visible photons into UV emission. (2) Integrating CSNPs with CdS NPs to construct core–shell structured CSNPs@CdS, expanding CdS’s light absorption range. (3) Introducing Au NPs to form four-layer core–shell structured CSNPs@CdS@Au, enhancing CdS stability and catalytic efficiency. Under 300 W xenon lamp irradiation, CSNPs@CdS@Au demonstrated exceptional photocatalytic performance and stability in RhB degradation. This study contributes to advancing the NIR photocatalytic activity of CdS-based photocatalysts.

## 2. Materials and Methods

### 2.1. Chemicals and Materials

Yttrium chloride hexahydrate (99.9%), ytterbium chloride hexahydrate (99.9%), erbium chloride hexahydrate (99.9%), ammonium fluoride (AR), n-hexane (99%), sodium hydroxide (AR), sodium oleate (97.0%), oleic acid (OA, 90%), trisodium citrate dihydrate (98%), 3-aminopropyltriethoxysilane, and ammonia aqueous solution (25–28 wt%) were purchased from Aladdin Bio-Chemistry Shanghai China Technology Co., Ltd. (Shanghai, China), poly (vinylpyrrolidone) (PVP, MW = 55,000), Thioacetamide (TAA, 99.0%), cadmium chloride (99%) and 1-octadecene (ODE, 90% technical grade) were purchased from Beijing Innochem Technology Co., Ltd. (Beijing, China). Rh B and chloroauric acid (99.9%) were purchased from Anhui Zaisheng Technology Co., Ltd. (Hefei, China), and Cyclohexane (99.7%) was purchased from Hunan Huihong Reagent Co., Ltd. (Changsha, China). Deionized (DI) water was used during the entire experimental process. All chemical reagents were used without any purification.

### 2.2. Synthesis of CSNPs@CdS@Au NPs

As shown in Figure 1, the synthesis of CSNPs@CdS@Au four-layer core–shell structure NPs is achieved using a four-step process. Firstly, lanthanide oleate complexes were used as precursors to synthesize NaYF_4_:Yb^3+^,Er^3+^ core NPs by thermal decomposition method. Then, NaYF_4_ inert shell layer was coated on the surface of NaYF_4_:Yb^3+^,Er^3+^ core to obtain CSNPs. Subsequently, CdS was encapsulated on the surface of CSNPs through ion deposition, and finally, gold was encapsulated on the surface of CdS from chloroauric acid through redox reaction.

#### 2.2.1. Synthesis of NaYF_4_:Yb^3+^,Er^3+^ Core NPs

RE fluoride NPs were prepared via thermal decomposition [36]. Into a three-necked flask containing a solvent mixture of DI water (6 mL), ethanol (7 mL), and n-hexane (14 mL), 0.4732 g (1.56 mmol) YCl_3_ 6H_2_O, 0.1549 g (0.40 mmol) YbCl_3_ 6H_2_O, 0.0158 g (0.04 mmol) ErCl_3_ 6H_2_O, and 1.8570 g (6.1 mmol) sodium oleate were sequentially added. The molar ratio of Y:Yb:Er was set at 78:20:2. The solution was stirred at room temperature for 0.5 h and then transformed to a preheated water bath of 70 °C. The whole reaction system was maintained at 70 °C for 4 h with vigorous stirring. After the completion of the reaction, the upper oil phase containing Y_0.78_Yb_0.20_Er_0.02_-oleate complexes were washed three times with DI water (20 mL × 3) to remove the byproduct of NaCl. The above prepared Y_0.78_Yb_0.20_Er_0.02_-oleate complexes (2 mmol) were mixed with 12 mL OA and 30 mL ODE in a three-necked flask at room temperature with vigorous stirring. The solution was degassed and heated to 140 °C to form a transparent solution, and then cooled down to 50 °C. Subsequently, a methanol solution (20 mL) containing NaOH (0.2000 g, 5.0 mmol) and NH_4_F (0.2963 g, 8.0 mmol) was added to a flask and stirred continuously for 1.5 h. After that, methanol was removed by heating the solution at 100 °C. The reaction mixture was then heated to 310 °C and maintained for 1.5 h under a N_2_ atmosphere. After the completion of the reaction, the reaction system was cooled down to room temperature and the resultant solution was precipitated in 40 mL ethanol. The CNPs were collected by centrifugation (11,000 rpm) and washed five times with a mixture of cyclohexane and ethanol at a volume ratio of 1:3.

#### 2.2.2. Synthesis of NaYF_4_:Yb^3+^, Er^3+^@NaYF_4_ Core-Inert Shell NPs

Into a flask containing a solvent mixture of DI water (6 mL), ethanol (7 mL) and *n*-hexane (14 mL), 0.6068 g (2 mmol) YCl_3_ 6H_2_O and 1.8570 g (6.1 mmol) sodium oleate were sequentially added. The solution was stirred at 70 °C for 4 h. After the complement of the reaction naturally cooled to room temperature, the upper oil phase was washed three times with DI water (20 mL × 3) to remove the byproduct of NaCl. The supernatant was poured into a 100 mL three-necked flask and mixed with 12 mL OA and 30 mL ODE. The mixed solution was heated to 140 °C and continuously stirred to form a transparent solution, and then cooled down to 70 °C; subsequently, 4 mL of cyclohexane containing 0.3 g CNPs were added to the mixture solution and stirred for 5 min. After that, the cyclohexane was removed by maintaining the mixture solution at 70 °C under vacuum for 30 min, and then the solution was cooled down to 50 °C. Meanwhile, a methanol solution (20 mL) containing NaOH (0.2000 g, 5.0 mmol) and NH_4_F (0.2963 g, 8.0 mmol) was added to the reaction flask, and stirred continuously for 1.5 h. After that, methanol was removed by heating the solution at 100 °C. The reaction mixture was then heated to 310 °C and maintained for 1.5 h under N_2_ atmosphere. Then, the reaction system was cooled down to room temperature and the resultant solution was precipitated in 40 mL ethanol. The CSNPs were collected by centrifugation (11,000 rpm) and washed five times with a mixture of cyclohexane and ethanol at a volume ratio of 1:3.

#### 2.2.3. Synthesis of CSNPs@CdS Three-Layer Core–Shell Structural Composite

Sandwich-structure NPs were prepared by the ion deposition method [26]. A quantity of 0.1200 g CSNPs was dispersed in 140 mL DI water by sonication treatment for 30 min. Trisodium citrate dihydrate (7 mL, 0.1 M) and CdCl_2_ (7 mL, 0.08 M) were sequentially added to the above solution under continuous stirring for 20 min and 40 min, respectively. After that, ammonia aqueous solution was slowly dropped into the mixture solution until the pH value reached 10.5. Subsequently, the mixture solution was slowly heated to 65 °C in a bath. Then, 10 mL TAA (0.063 M) was added into the above solution at a rate of 0.1 mL/min. Stirring was maintained at 65 °C for 1 h. The final product was washed with ethanol and DI water three times and dried at 60 °C for 12 h in an oven.

#### 2.2.4. Synthesis of CSNPs@CdS@Au Four-Layer Core–Shell Structural Composites

Au NPs were loaded onto the surface of CSNPs@CdS particles [34]. CSNPs@CdS 30 mg was dispersed in 5 mL cyclohexane and sonicated for 30 min. PVP (30 mg) and 30 mL ethanol solution (*v*/*v* ratio of water/ethanol = 9:1) were then added to the above solution for 30 min. Then, 100 μL 0.6 mM HAuCl_4_·4H_2_O was added to the above solution, and the reaction was stirred for 60 min. NaBH_4_ (0.0057 g) was added under 0 °C for 1 h and freeze-dried to obtain the final product CSNPs@CdS@Au. x represents the volume of 0.6 mM HAuCl_4_·4H_2_O added in μL, the amount of Au added is denoted as Au_x_, and the final product is denoted as CSNPs@CdS@Au_x_. For example, Au_100_ indicates that 100 μL of 0.6 mM HAuCl_4_·4H_2_O solution was added during the synthesis of CSNPs@CdS@Au.

### 2.3. Characterization

X-ray powder diffractometer (XRD) was used to analyze the purity and phase structure of the prepared materials using a Bruker D8 Advance model from Germany in the 2θ range from 10° to 80° at a scan rate of 2°/min. A transmission electron microscope (TEM) was used to characterize the size and morphology of the prepared materials using a JEOL JEM-F200 model from Tokyo, Japan. X-ray photoelectron spectroscopy (XPS) was used to analyze the elemental composition and the chemical state of the surface of the prepared material samples, using the Thermo Scientific K-Alpha model from the Waltham, MA, USA. A PL fluorescence spectrometer was used (Edinburgh FLS1000, Manchester, UK). The absorbance curve of the solution was measured using a UV-Visible spectrophotometer (TU1810), with the wavelength range being 200–800 nm.

### 2.4. Photocatalytic Activity Measurement

The photocatalytic properties of the prepared composites (CSNPs@CdS@Au) were evaluated by degradation of RhB solution under irradiation of a 300 W xenon lamp. The visible band (λ > 420 nm) was obtained by using a filter. In each experiment, 25 mg of the photocatalyst was mixed with 50 mL of RhB (1.0 × 10^−4^ M). Prior to light irradiation, the suspension was magnetically stirred in the dark for 30 min to achieve adsorption–desorption equilibrium [37,38,39]. During light irradiation, aliquots of approximately 2.5 mL were drawn at quantitative time intervals, filtered through a 0.22 μm polyethersulfone (PES) membrane to remove the photocatalyst particles, and then the concentration of RhB remaining in the supernatant was detected by a UV-visible spectrophotometer (TU1810).

## 3. Results and Discussion

### 3.1. X-Ray Diffraction (XRD) Analysis

The XRD of CSNPs, CSNPs@CdS, and CSNPs@CdS@Au are shown in Figure 1. The black line in Figure 1a represents the XRD plot of CSNPs, the diffraction peaks appearing at diffraction angles of 17.20°, 30.06°, 30.78°, 39.67°, 43.49°, and 53.27° correspond to the diffraction peaks of NaYF_4_ (100), (110), (101), (111), (201), and (300), respectively, and by comparing with the standard diffractogram of hexagonal phase (JCPDS#28-1192) [40], it is possible to determine that the synthesized CSNPs is a hexagonal crystal phase [15]. Moreover, no other impurity peaks are observed in this figure, and the strong and sharp peaks indicate that the synthesized CSNPs nanomaterials under thermal decomposition conditions are highly crystalline. The blue line Figure 1b shows the XRD plots of CSNPs@CdS NPs, with the crystallographic planes corresponding to the diffraction peaks appearing at diffraction angles of 26.50°, 43.96°, 52.13°, 54.58°, and 64.02° as CdS (111), (220), (311), (222), and (400), respectively. The successful preparation of CdS particles can be determined by comparison with the CdS standard PDF card (JCPDS#10-0454). The red line in Figure 1c shows the XRD patterns of CSNPs@CdS@Au NPs, with diffraction peaks detected at diffraction angles of 38.18° and 44.39°, which correspond to the (111) and (200) planes of metallic Au (JCPDS#04-0784) [41], confirming the successful loading of Au NPs.

### 3.2. TEM Analysis

In order to investigate the microscopic morphological and structural changes in the composite nanomaterials, TEM tests were performed on a series of samples. Figure 2a shows the CNPs prepared by the thermal decomposition method. The CNPs have a uniform particle morphology with an average particle size of approximately 21.7 nm (inset of Figure 2a), and the lattice stripe spacing is 0.32 nm (Figure 2b), corresponding to the NaYF_4_ (110) crystal plane. In order to perform the fluorescence enhancement of the NPs, a layer of NaYF_4_ inert shell (not doped) was encapsulated on top of CNPs; the encapsulation of the inert shell layer reduces the surface defects of the Yb^3+^, Er^3+^-doped active nuclei, as shown in Figure 2c. The synthesized CSNPs NPs are uniform and well-dispersed, with a smooth hexagonal surface morphology. The average particle size is approximately 52.8 nm (inset of Figure 2c), and the lattice stripe spacing is 0.32 nm (Figure 2d), corresponding to the NaYF_4_ (110) crystal plane [42,43,44]. The core–shell CSNPs@CdS NPs were successfully synthesized via a redox method (Figure 2e). It can be seen from the image that the average particle size is approximately 77 nm (inset of Figure 2e). Furthermore, the elemental mapping of CSNPs@CdS (Figure 3) clearly shows that S and Cd are distributed in the shell layer, while the elements constituting the CSNPs core are concentrated in the center. Through the linear scanning plot of Figure 3, it can be observed that the S and Cd distributions show an elevated, then decreased, and then increased situation, which proves that we successfully prepared CSNPs@CdS with a core–shell structure. The CdS NPs were completely and uniformly coated on the surface of the CSNPs, enabling them to absorb the excitation light of the CSNPs while maintaining stability [45]. The measured spacings from the lattice fringes of CdS were found to be 0.33 and 0.20 nm by HR-TEM Figure 2f, which correspond to the (111) and (220) crystallographic planes of CdS, respectively. The TEM of the CSNPs@CdS@Au is shown in Figure 2g, and the morphology of the material is that of spherical particles of uniform size, The average particle size is about 84 nm (inset of Figure 2g). Comparing with Figure 2e, it can be clearly seen that the introduction of Au makes the surface of the coated CdS particles smoother. Figure 2h shows that the spacing of the two lattice fringes is 0.23 and 0.20 nm, respectively, corresponding to the (111) and (200) crystal planes of Au [46]. Figure 2i is the elemental mapping diagram of CSNPs@CdS@Au particles. It can be observed from the figure that elements such as F, Na, Y, Er, and Yb are distributed in the core center of the particle, S and Cd elements are distributed on the outer layer of the CSNPs, and Au element is evenly distributed on the outermost layer. The above results further indicate the successful preparation of the four core–shell composite material. Figure 4a confirms the presence of F, Na, Y, Yb, and Er elements in the CSNPs, with an Yb:Er molar ratio of 20:2, consistent with the expected composition. Figure 4b also confirms the presence of S, Cd, and Au elements, further confirming the successful preparation of the quad-core–shell composite material.

### 3.3. XPS Analysis

The chemical states of the involved elements of CSNPs@CdS@Au were investigated by XPS. The survey spectrum of CSNPs@CdS@Au reveals the co-presence of Cd, S, Na, Y, F, and Au elements (Figure 5a). Due to their relatively low doping concentrations (2% for Er^3+^ and 20% for Yb^3+^) and the shielding effect of the outer CdS and Au layers, the survey spectrum did not reveal the presence of Yb and Er. All XPS spectra were calibrated using the adventitious carbon C 1s (Figure 5b) band at 284.8 eV. The core energy spectrum of Cd exhibits two spin–orbit bands at approximately 411.48 eV and 404.78 eV, as shown in Figure 5c, which correspond to Cd 3d_3/2_ and 3d_5/2_, respectively. This confirms the oxidation state of CdS compounds as +2. And the high-resolution spectrum of S 2p (Figure 5d) exhibits two bands at 161.18 eV (S 2p_1/2_) and 159.08 eV (S 2p_3/2_) that can be assigned to S^2−^. The Cd and S element spectrum in CSNPs@CdS@Au material shows a shift compared with that in CSNPs@CdS. When the CdS shell wraps around Au NPs, the close contact between CdS and Au leads to a SPR effect [46], causing electrons to transfer from Au to CdS and inhibiting the recombination of electron–hole pairs [47]. The Au 4f spectra (Figure 5e) show the zero-oxidation state of Au 4f_7/2_ bands at 83.38 and 87.98 eV, and the peak of Au 4f_5/2_ bands at 84.78 and 92.68 eV, which may be due to the charge transfer induced by the Au NPs on the CdS carrier [48]. This indicates that the introduction of Au has caused an electron shift in CdS. In addition, the high-resolution spectra of Y 3d, Yb 4d, and Er 4d are obviously found in Figure 5f–h.

### 3.4. Photoluminescence (PL) Spectra Analysis

Figure 6a shows the UC PL spectra of CNPs, CSNPs, CSNPs@CdS, and CSNPs@CdS@Au, respectively, under excitation at 980 nm. It is well known that NaYF_4_ particles are an excellent host matrix for UC luminescence when co-doped with Yb^3+^ and Er^3+^ in NaYF_4_. Under 980 nm NIR light excitation, CNPs produces strong UC emission, with Yb^3+^ acting as a sensitizing ion for the sequential absorption of 980 nm NIR light and Er^3+^ acting as an activating ion for the emission of a series of spectra at wavelengths smaller than 980 nm [44]. There are three distinct bands at 520, 540, and 654 nm attributed to the ^2^H_11/2_→^4^I_15/2_, ^4^S_3/2_→^4^I_15/2_, and ^4^F_9/2_→^4^I_15/2_ transition, and a weak peak at 407 nm, which is usually attributed to the ^2^H_9/2_→^4^I_15/2_ leap of the Er^3+^ ion [42]. The higher emission intensity of the CSNPs core shell than that of the CNPs core is observed from the green line in the figure, which originates from the suppression of CNPs surface quenching. After coating the CSNPs, the fluorescence intensities of CSNPs@CdS and CSNPs@CdS@Au decreased significantly. By integrating the UV emission bands of the UC emission spectra of the four materials, as shown in Figure 6b, under the excited light of 980 nm, the UV emission intensity of CSNPs@CdS@Au is 3.19% of that of CSNPs. This shows that the CdS shell in CSNPs@CdS@Au effectively absorbs more than 96.81% of the UV emission. This shows that the unique four-layer core–shell structure of CSNPs@CdS@Au can greatly enhance the light absorption.

### 3.5. Photocatalytic Performance

To evaluate the photocatalytic performance of CSNPs@CdS@Au in the degradation of RhB, we compared it with CSNPs, CdS, CSNPs@CdS, and a physical mixture of CSNPs and CdS (CSNPs/CdS). A blank experiment without any catalyst was also conducted for reference. It is reported that RhB has a maximum absorption peak at a wavelength of 554 nm [49]. Figure 7a shows the time-dependent photocatalytic degradation of RhB by CSNPs@CdS@Au.

As shown in Figure 7b, the catalytic experiments of different catalysts are compared, and it can be seen that the orange line represents the catalytic rate of bare CdS. After 15 min of catalysis by bare CdS, the degradation efficiency of RhB is approximately 9.8% [degradation efficiency = (1 − *C*/*C*_0_) × 100%, where *C*_0_ is the initial concentration and *C* is the post-irradiation concentration]. The black line represents CSNPs. The degradation efficiency of RhB within 15 min is approximately 5.0%, which is close to the adsorption effect, indicating that CSNPs primarily function as light converters to emit UV/vis light but lack inherent catalytic activity for RhB degradation by themselves. The green line represents the physical mixture catalyst composed of CdS and CSNPs. The degradation rate after 15 min is 6.9%, indicating that the physical mixture of the two does not enhance the catalytic efficiency. The blue line represents the catalytic rate of CSNPs@CdS. Within 15 min, it degraded 81% of RhB, indicating that the core–shell structure of CSNPs@CdS NPs has high catalytic performance. The CSNPs@CdS@Au catalyst (red line) demonstrated the highest activity, degrading 97.7% of RhB within 15 min. The incorporation of Au NPs significantly enhanced the performance compared to CSNPs@CdS. It can be seen that the four-layer core–shell structure of the CSNPs@CdS@Au catalyst, due to its structural specificity, has a superior performance compared with other catalysts, as well as catalysts of other materials reported previously [18,26,50,51].

Correspondingly, to further quantify the degradability of this series of samples, we performed the calculation of the first-order kinetic fitting constant *k* value using the equation below: −ln (*C*/*C*_0_) = *kt*. Where *t* represents irradiation time, *k* represents the pseudo-first-order kinetic constant, *C*_0_ represents the initial concentration of RhB, and *C* represents RhB concentration at a moment. From Figure 7c,d, The CSNPs@CdS@Au exhibit the highest *k* value (0.2745 min^−1^), which is significantly higher than CSNPs@Cds (0.1018 min^−1^), CSNPs/Cds (0.0067 min^−1^), and bare Cds (0.0071 min^−1^). As shown in Figure 7e, by controlling the content of Au element as the variable, the optimal ratio of Au loading onto CSNPs@Cds was obtained. When the content of added Au was lower than 100 μL, with the increase in Au content, the degradation efficiency in 15 min was 95.8% for the catalyst with the dosage of Au_10_, 96.2% for Au_30_, and 96.6% for Au_50_, which indicated that the higher the dosage of catalyst dosed, the higher the degradation rate of RhB. This may be due to an increase in the number of photocatalytic active sites (introduced by Au NPs), hence more reactive radicals being produced with higher Au loading (up to Au_100_). However, when the Au content exceeded 100 μL, the catalytic efficiency actually decreased. This might be due to the excessive increase in Au content, which occupied the active sites of CdS, resulting in a decrease in the catalytic rate. Correspondingly, cycling photocatalytic reactions were also performed on CSNPs@CdS@Au to verify the recycling ability of CSNPs@CdS@Au. From Figure 7f and Table 1, the CSNPs@CdS@Au demonstrates great recyclability because the photodegradation efficiency of RhB is still kept at 93.25% after five cycles. As a result of their excellent structural stability, CSNPs@CdS@Au possess high reliability. Table 2 presents the comparative results between the CdS-based catalyst and similar photocatalysts previously reported for RhB degradation.

### 3.6. Photocatalytic Mechanism

The active species of CSNPs@CdS@Au during the photocatalytic reaction were identified via trapping experiments. Various scavengers, including triethanolamine (TEOA), 1,4-Benzoquinone (p-BQ), and isopropyl alcohol (IPA), were introduced into the solution of RhB [34,53]. The h^+^, O_2_^−^ and OH were trapped by TEOA, p-BQ, and IPA, respectively [54]. As shown in Figure 8a–d, the addition of p-BQ, TEOA, and IPA all resulted in a significant decrease in the catalytic rate. After 15 min of reaction, the degradation rates were 46.52% for p-BQ, 20.86% for TEOA, and 89.68% for IPA. The change in degradation rate was particularly pronounced when TEOA was introduced into the reaction system. It can therefore be inferred that the O_2_^−^ serves as the dominant active species responsible for RhB degradation over this catalyst, while the e^−^ acts as a secondary contributor. It is noted that RhB dye itself could act as a photosensitizer in the initial stage of the photocatalytic process. However, the active species trapping experiments and the significant performance difference between catalysts (e.g., bare CdS vs. CSNPs@CdS@Au) strongly suggest that the degradation is predominantly driven by the catalysts, where the UC process and the Au/CdS Schottky junction play decisive roles in generating active radicals (mainly O_2_^−^).

Meanwhile, Figure 8e displays the photocurrent response curves of various photocatalysts, which alternate with the on-off cycles of the lamp. Among them, the CSNPs@CdS@Au sample exhibits the highest transient photocurrent response, indicating superior photo-generated electron–hole transfer efficiency compared to other samples—typically signifying enhanced photocatalytic activity. The gradual increase in photocurrent intensity during the on-light intervals for the CSNPs@CdS, CSNPs, and CNPs samples may involve multiple factors: First, the slow rise in photocurrent in semiconductors (known as the relaxation phenomenon) reveals the semiconductor’s sensitivity to light intensity. The Au/CdS heterojunction formed by morphological differences alters carrier flow [33]. Second, it may be influenced by the synergistic effects between carrier flow direction and effective defects. Additionally, the decrease in photocurrent intensity in CSNPs@CdS@Au can be attributed to hole accumulation. To further validate the superiority of CSNPs@CdS@Au over other structured photocatalysts in terms of carrier mobility, electrochemical impedance spectroscopy (EIS) measurements were conducted (Figure 8f). Compared to other samples, CSNPs@CdS@Au exhibited a significantly smaller circular measurement radius. Furthermore, fitting the EIS curves yielded charge transfer resistance values of 8.59 Ω, 9.182 Ω, 12.76 Ω, and 13.33 Ω for CSNPs@CdS@Au, CSNPs@CdS, CSNPs, and CNPs, respectively. The results indicate that CSNPs@CdS@Au exhibits more efficient surface charge transfer capability, thereby achieving optimal photocatalytic activity, further confirming the superiority of the CSNPs@CdS@Au morphology.

Under 980 nm laser irradiation, Yb^3+^ absorbs near-infrared photons and makes a ^2^F_7/2_→^2^F_5/2_ energy transition, which then transitions to the ground state ^2^F_7/2_ and transfers energy to nearby Er^3+^ [36]. As shown in Figure 9, after absorbing energy, the electrons of Er^3+^ can reach the excited state through the transition of ^4^I_15/2_→^4^I_11/2_, and the electrons in the excited state can transition to the excited state of a higher energy level through the transition of ^4^I_11/2_→^4^F_7/2_ or ^4^I_13/2_-^4^F_9/2_ due to energy matching [36]. Er^3+^-released energy through non-radiative relaxation can transition to ^2^H_11/2_ and ^4^S_3/2_ orbital energy levels [57], and when the electrons in these two orbitals transition back to the ground state, they radiate light at 522 and 541 nm, respectively. On the other hand, a ^4^F_9/2_→^4^I_15/2_ transition from a higher energy level to an electron in a ^4^F_9/2_ orbital through non-radiative relaxation or from an ^4^I_13/2_ orbital to a ^4^F_9/2_ orbital will radiate 654 nm of light. However, this emitted light wave energy can be absorbed by the catalyst and then produce electron–hole pairs to decompose RhB. These photogenerated carriers are the core of the photocatalytic reaction and have strong Redox ability, which can trigger a series of Redox reactions.

## 4. Conclusions

In summary, we successfully developed a four-layer core–shell structured photocatalyst (CSNPs@CdS@Au). Experimental results demonstrate that the composite catalyst exhibits remarkable enhancement in both catalytic activity under visible light and cyclic stability compared to other catalysts. The performance enhancement mechanism can be attributed to three key factors: (1) the uniquely designed multilayer core–shell architecture significantly enhances structural stability through physical isolation effects; (2) the NaYF_4_ inert shell encapsulation amplifies UC luminescence intensity by suppressing surface quenching; and (3) the Schottky junction formed between Au NPs and CdS effectively promotes spatial separation of photogenerated charge carriers and increases reactive active sites. In addition, further analysis suggests that optimizing the quantum yield of UC materials (e.g., via optimizing lanthanide doping) or developing Z-scheme heterojunction configurations could unlock new frontiers in photocatalytic efficiency. This work not only provides crucial theoretical foundations and practical references for constructing efficient and stable photocatalytic systems but also opens new avenues for designing advanced catalysts for full-spectrum solar energy utilization.

## Data Availability

The original contributions presented in this study are included in the article. Further inquiries can be directed to the corresponding authors.

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
