# Peer review of "Unique Four-Layer Core–Shell NaYF4:Yb3+,Er3+@NaYF4@CdS@Au Nanocomposites for Enhanced Full-Spectrum Photocatalytic Degradation of Rhodamine B"

_molecules, 2025, doi:10.3390/molecules30214215_

Round 1

Reviewer 1 Report

Comments and Suggestions for Authors
  • The incorporation of rare earth elements (e.g., Y, Yb, Er) in the synthesis of the NaYF4:Yb³⁺, Er³⁺@NaYF4@CdS@Au nanocomposite system raises concerns regarding scalability and industrial viability. Rare earth metals are expensive, geographically limited, and environmentally challenging to extract and process. 
  • The reported complexity of the nanocomposite synthesis process may hinder reproducibility, a critical factor for scientific validation and potential industrial adoption. Please include a step-by-step synthesis protocol in the supplementary information, specifying the reaction conditions (e.g., temperature, time, precursor ratios, and Au content).
  • The manuscript would benefit from a comparative study of the photocatalytic performance of the NaYF4:Yb³⁺, Er³⁺@NaYF4@CdS@Au system against other well-established upconversion materials for the same target reaction.
  • The authors propose that charge transfer occurs from gold nanoparticles (Au NPs) to cadmium sulfide (CdS), but this claim lacks experimental validation. To substantiate this mechanism, include transient photocurrent response or electrochemical impedance spectroscopy (EIS) data to demonstrate charge transfer dynamics between Au NPs and CdS.
  • The arguments presented in Section 3.6 (Active Species Trapping Experiments) cannot be critically evaluated due to the absence of supplementary data.

Author Response

Dear Reviewer:

Thank you for reviewing our manuscript “Unique Four-Layer Core-Shell NaYF4:Yb³⁺,Er³⁺@NaYF4@CdS@Au Nanocomposites for Enhanced Full-Spectrum Photocatalytic Degradation of Rhodamine B” (Ms. Ref. No.: molecules-3891318) and providing constructive suggestion which have strengthened our manuscript. We have now carefully revised the manuscript according to the comment.

Review 1:

Comment 1-1: The incorporation of rare earth elements (e.g., Y, Yb, Er) in the synthesis of the NaYF4:Yb3+, Er3+@NaYF4@CdS@Au nanocomposite system raises concerns regarding scalability and industrial viability. Rare earth metals are expensive, geographically limited, and environmentally challenging to extract and process.

Response: We thank the reviewer for the kind advice. We fully understand and appreciate this important point raised by the reviewers. The use of rare earth elements is indeed a consideration for large-scale applications in the field of advanced functional materials. The primary objective of this study is to explore and validate the scientific feasibility of a novel multi-layer core-shell structural design concept for enhancing photocatalytic performance. We believe this structural design strategy possesses inherent universality and can be explored for future application in catalytic systems based on more affordable and abundant elements (such as upconversion composites based on TiO2, ZnO, etc.). We have incorporated relevant discussion in the paper's conclusion section, explicitly outlining this future research direction to reflect our commitment to addressing this issue. We also express our gratitude to the reviewers for their positive feedback, which provides us with important reference suggestions for our subsequent research.

Comment 1-2: The reported complexity of the nanocomposite synthesis process may hinder reproducibility, a critical factor for scientific validation and potential industrial adoption. Please include a step-by-step synthesis protocol in the supplementary information, specifying the reaction conditions (e.g., temperature, time, precursor ratios, and Au content).

Response: We thank the reviewer for the kind advice. We have carefully revised the manuscript according to your suggestion, please review it again. To ensure experimental reproducibility, we have detailed the specific conditions for each synthetic reaction step in Section 2.2 of the manuscript, including temperature, duration, precursor ratios, and other parameters. Additionally, we have clarified that the loading amount of gold nanoparticles is based on the volume of the added chloroauric acid solution (Aux, e.g., Au~100~ denotes the addition of 100 μL of 0.6 mM HAuCl₄). This has been more clearly explained in Section 2.2.4 of the manuscript.

Before revised:

“2.2.1 Synthesis of NaYF4:Yb3+, Er3+ core NPs

Preparation of RE fluoride NPs by thermal decomposition. Into a three-necked flask containing a solvent mixture of DI water (6 mL), ethanol (7 mL) and n-hexane (14 mL), 0.4732 g (1.56 mmol) YCl3⋅6H2O, 0.1549 g (0.40 mmol) YbCl3⋅6H2O, 0.0158 g (0.04 mmol) ErCl3⋅6H2O and 1.8570 g (6.1 mmol) sodium oleate were sequentially added. The molar ratio of Y:Yb:Er was set at 78:20:2. The solution was stirred at room temperature for 0.5 h and then transformed to a preheated water-bath of 70 ℃. The whole reaction system was maintained at 70 ℃ for 4 h with vigorous stirring. After the completion of the reaction, the upper oil phase containing Y0.78Yb0.20Er0.02-oleate complexes were washed three times with DI water (20 mL×3) to remove the byproduct of NaCl. Subsequently, n-hexane was evaporated off under vacuum, resulting in Y0.78Yb0.20Er0.02-oleate complexes as a waxy solid. The above prepared Y0.78Yb0.20Er0.02-oleate complexes (2 mmol) were mixed with 12 mL OA and 30 mL ODE in a three-necked flask at room temperature with vigorous stirring. The solution was degassed, and heated to 140 ℃ to form a transparent solution, and then cooled down to 50 ℃. Subsequently, a methanol solution (20 mL) containing NaOH (0.2000 g, 5.0 mmol) and NH4F (0.2963 g, 8.0 mmol) was added to flask and stirred continuously for 1.5 h the formed colloidal solution was stirred for 1 h. After that, methanol was removed by heating the solution at 100 ℃. The reaction mixture was then heated to 310 ℃ and maintained for 1.5 h under N2 atmosphere. After the completion of the reaction, the reaction system was cooled down to room temperature and the resultant solution was precipitated in 40 mL ethanol. The CNPs were collected by centrifugation (11000 rpm) and washed five times with a mixture of cyclohexane and ethanol at a volume ratio of 1:3.

2.2.2 Synthesis of NaYF4:Yb3+, Er3+@NaYF4 core- inert shell NPs

Into a flask containing a solvent mixture of DI water (6 mL), ethanol (7 mL) and n-hexane (14 mL), 0.6068 g (2 mmol) YCl3⋅6H2O, and 1.8570 g (6.1 mmol) sodium oleate were sequentially added. The solution was stirred at 70 ℃ for 4 h. After the complement of the reaction, naturally cooled to room temperature, the upper oil phase was washed three times with DI water (20 mL×3) to remove the byproduct of NaCl. The supernatant was poured into a 100 mL three-necked flask, and mixed with 12 mL OA and 30 mL ODE. The mixed solution was heated to 140 ℃ and continuously stirred to form a transparent solution, and then cooled down to 70 ℃, subsequently, 4 mL of cyclohexane containing 0.3 g CNPs were added in the mixture solution and stirred for 5 min. After that the cyclohexane was removed by maintaining the mixture solution at 70 ℃ under vacuum for 30 min and then the solution was cooled down to 50 ℃. Meanwhile, a methanol solution (20 mL) containing NaOH (0.2000 g, 5.0 mmol) and NH4F (0.2963 g, 8.0 mmol) was added to the reaction flask, and stirred continuously for 1.5 h. After that, methanol was removed by heating the solution at 100 ℃. The reaction mixture was then heated to 310 ℃ and maintained for 1.5 h under N2 atmosphere. Then the reaction system was cooled down to room temperature and the resultant solution was precipitated in 40 mL ethanol. The CSNPs were collected by centrifugation (11000 rpm) and washed five times with a mixture of cyclohexane and ethanol at a volume ratio of 1:3.

2.2.3 Synthesis of CSNPs@CdS three-layer core-shell structural composite

Sandwich structure NPs were prepared by ion deposition method[26]. 0.1200 g CSNPs was dispersed in 140 mL DI water by sonication treatment for 30 min. Trisodium citrate dihydrate (7 mL, 0.1 M) and CdCl2 (7 mL, 0.08 M) were sequentially added to the above solution under continuous stirring for 20 min and 40 min, respectively. After then, ammonia aqueous solution was slowly dropped in the mixture solution until the pH value reached 10.5. Subsequently, the mixture solution was slowly heated to 65 ℃ in a bath. Then, 10 mL TAA (0.063 M) was added into the above solution at a rate of 0.1 mL/min. Stirring was maintained at 65 ℃ for 1 h. The final product was washed with ethanol and DI water for three times, and dried at 60 ℃ for 12 h in an oven.

2.2.4 Synthesis of CSNPs@CdS@Au four-layer core-shell structural composites

Au NPs was loaded onto the surface of CSNPs@CdS particles[34]. CSNPs@CdS 30 mg was dispersed in 5 mL cyclohexane and sonicated for 30 min. 30 mg PVP and 30 mL ethanol solution (v/v ratio of water:ethanol=9:1) were then added to the above solution for 30 min. 100 μL 0.6 mM HAuCl4·4H2O was then added to the above solution, and the reaction was stirred for 60 min. 0.0057 g NaBH4 was added under 0 ℃ for 1 h, and freeze-dried to obtain the final product CSNPs@CdS@Au. x represents the volume of 0.6 mM HAuCl4·4H2O added in μL, the amount of Au added is denoted as Aux, and the final product is denoted as CSNPs@CdS@Aux.”

After revised:

“2.2.1 Synthesis of NaYF4:Yb3+, Er3+ core NPs

Preparation of RE fluoride NPs by thermal decomposition[36]. Into a three-necked flask containing a solvent mixture of DI water (6 mL), ethanol (7 mL) and n-hexane (14 mL), 0.4732 g (1.56 mmol) YCl3⋅6H2O, 0.1549 g (0.40 mmol) YbCl3⋅6H2O, 0.0158 g (0.04 mmol) ErCl3⋅6H2O and 1.8570 g (6.1 mmol) sodium oleate were sequentially added. The molar ratio of Y:Yb:Er was set at 78:20:2. The solution was stirred at room temperature for 0.5 h and then transformed to a preheated water-bath of 70 ℃. The whole reaction system was maintained at 70 ℃ for 4 h with vigorous stirring. After the completion of the reaction, the upper oil phase containing Y0.78Yb0.20Er0.02-oleate complexes were washed three times with DI water (20 mL×3) to remove the byproduct of NaCl. Subsequently, n-hexane was evaporated off under vacuum, resulting in Y0.78Yb0.20Er0.02-oleate complexes as a waxy solid. The above prepared Y0.78Yb0.20Er0.02-oleate complexes (2 mmol) were mixed with 12 mL OA and 30 mL ODE in a three-necked flask at room temperature with vigorous stirring. The solution was degassed, and heated to 140 ℃ to form a transparent solution, and then cooled down to 50 ℃. Subsequently, a methanol solution (20 mL) containing NaOH (0.2000 g, 5.0 mmol) and NH4F (0.2963 g, 8.0 mmol) was added to flask and stirred continuously for 1.5 h the formed colloidal solution was stirred for 1 h. After that, methanol was removed by heating the solution at 100 ℃. The reaction mixture was then heated to 310 ℃ and maintained for 1.5 h under N2 atmosphere. After the completion of the reaction, the reaction system was cooled down to room temperature and the resultant solution was precipitated in 40 mL ethanol. The CNPs were collected by centrifugation (11000 rpm) and washed five times with a mixture of cyclohexane and ethanol at a volume ratio of 1:3.

2.2.2 Synthesis of NaYF4:Yb3+, Er3+@NaYF4 core- inert shell NPs

Into a flask containing a solvent mixture of DI water (6 mL), ethanol (7 mL) and n-hexane (14 mL), 0.6068 g (2 mmol) YCl3⋅6H2O, and 1.8570 g (6.1 mmol) sodium oleate were sequentially added. The solution was stirred at 70 ℃ for 4 h. After the complement of the reaction, naturally cooled to room temperature, the upper oil phase was washed three times with DI water (20 mL×3) to remove the byproduct of NaCl. The supernatant was poured into a 100 mL three-necked flask, and mixed with 12 mL OA and 30 mL ODE. The mixed solution was heated to 140 ℃ and continuously stirred to form a transparent solution, and then cooled down to 70 ℃, subsequently, 4 mL of cyclohexane containing 0.3 g CNPs were added in the mixture solution and stirred for 5 min. After that the cyclohexane was removed by maintaining the mixture solution at 70 ℃ under vacuum for 30 min and then the solution was cooled down to 50 ℃. Meanwhile, a methanol solution (20 mL) containing NaOH (0.2000 g, 5.0 mmol) and NH4F (0.2963 g, 8.0 mmol) was added to the reaction flask, and stirred continuously for 1.5 h. After that, methanol was removed by heating the solution at 100 ℃. The reaction mixture was then heated to 310 ℃ and maintained for 1.5 h under N2 atmosphere. Then the reaction system was cooled down to room temperature and the resultant solution was precipitated in 40 mL ethanol. The CSNPs were collected by centrifugation (11000 rpm) and washed five times with a mixture of cyclohexane and ethanol at a volume ratio of 1:3.

2.2.3 Synthesis of CSNPs@CdS three-layer core-shell structural composite

Sandwich structure NPs were prepared by ion deposition method[26]. 0.1200 g CSNPs was dispersed in 140 mL DI water by sonication treatment for 30 min. Trisodium citrate dihydrate (7 mL, 0.1 M) and CdCl2 (7 mL, 0.08 M) were sequentially added to the above solution under continuous stirring for 20 min and 40 min, respectively. After then, ammonia aqueous solution was slowly dropped in the mixture solution until the pH value reached 10.5. Subsequently, the mixture solution was slowly heated to 65 ℃ in a bath. Then, 10 mL TAA (0.063 M) was added into the above solution at a rate of 0.1 mL/min. Stirring was maintained at 65 ℃ for 1 h. The final product was washed with ethanol and DI water for three times, and dried at 60 ℃ for 12 h in an oven.

2.2.4 Synthesis of CSNPs@CdS@Au four-layer core-shell structural composites

Au NPs was loaded onto the surface of CSNPs@CdS particles[34]. CSNPs@CdS 30 mg was dispersed in 5 mL cyclohexane and sonicated for 30 min. 30 mg PVP and 30 mL ethanol solution (v/v ratio of water:ethanol=9:1) were then added to the above solution for 30 min. 100 μL 0.6 mM HAuCl4·4H2O was then added to the above solution, and the reaction was stirred for 60 min. 0.0057 g NaBH4 was added under 0 ℃ for 1 h, and freeze-dried to obtain the final product CSNPs@CdS@Au. x represents the volume of 0.6 mM HAuCl4·4H2O added in μL, the amount of Au added is denoted as Aux, and the final product is denoted as CSNPs@CdS@Aux. For example, Au100 indicates that 100 μL of 0.6 mM HAuCl4·4H2O solution was added during the synthesis of CSNPs@CdS@Au.”

Comment 1-3: The manuscript would benefit from a comparative study of the photocatalytic performance of the NaYF4:Yb3+,Er3+@NaYF4@CdS@Au system against other well-established upconversion materials for the same target reaction.

Response: We thank the reviewer for the kind advice. We have carefully revised the manuscript according to your suggestion, please review it again. We have conducted a broader review (Table 2) of recent literature, systematically summarizing and comparing the performance of various CdS-based and other upconversion material composite photocatalysts in RhB degradation. This table clearly demonstrates that the CSNPs@CdS@Au catalyst prepared in this work exhibits significant advantages over most reported catalysts in terms of reaction rate constants (k) and degradation efficiency. This strongly demonstrates the competitive performance of our uniquely designed quadruple-layer structure.

Table 2. Comparison of CdS-Based Catalysts for RhB Catalysis in Earlier Reports

Catalysts

k

Concentration

Deg. time

Deg. rate

 light source

Ref.

sodium alginate- CdS

0.0049 min-1

1 mg,

3 mL (10ppm)

60 min

94.98%

visible blue LED (12 W)

[10]

CdS/Mn-MOF(50)

0.0724 min-1

15 mg,

100 mL (10 mg/L)

10 min

60 min

55.7%,

98.7%

500 W Xe lamp

[52]

CeO2/CdS

0.0161 min-1

50 mg,

50 mL (20 mg/L)

180 min

94.5%

300 W Xe lamp

[53]

CdS/ZnO

0.063 min-1

50 mg,

50 mL (10 mg/L)

40 min

80 min

95%

99.55%

300 W Xe lamp

[54]

Au@CdS-CdS

0.0492 min-1

6 mg,

20 mL (1Í10-5 M)

60 min

45%

300 W Xe lamp

[47]

ZnO–CdS

0.0499 min-1

10 mg, 

100 mL (10 ppm)

80 min

98.16%

100 W solar simulator

[50]

CdS QDs/IO-TiO2

0.039 min-1

3 mg,

30 mL (50 mg/L)

50 min

85%

300 W Xe lamp

[14]

(α/β-CdS)/SiO2

0.044 min-1

50 mg,

30 mL (400 mg/L)

60 min

93.3%

50 mW/cm2 (113.7 mW)

[55]

CeCO3OH@(H/C–CdS)

0.0269 min-1

50 mg,

30 mL (2.000 g/L)

60 min

150 min

86.8%

99.6%

visible-light irradiation

[56]

CSNPs@CdS@Au

0.2745 min-1

25 mg,

50 mL (1Í10-4 M)

15 min

97.7%

300 W Xe lamp 

This work

Comment 1-4: The authors propose that charge transfer occurs from gold nanoparticles (Au NPs) to cadmium sulfide (CdS), but this claim lacks experimental validation. To substantiate this mechanism, include transient photocurrent response or electrochemical impedance spectroscopy (EIS) data to demonstrate charge transfer dynamics between Au NPs and CdS.

Response: We thank the reviewer for the kind advice. We have carefully revised the manuscript according to your suggestion, please review it again. We agree with the reviewer's point that direct electrochemical evidence would provide stronger support for charge transfer. The corresponding electrochemical testing was completed within the revision deadline. The data figures and analysis are presented in Section 3.6.

“Meanwhile, Fig. 8e displays the photocurrent response curves of various photocatalysts, which alternate with the on-off cycles of the lamp. Among them, the CSNPs@CdS@Au sample exhibits the highest transient photocurrent response, indicating superior photo-generated electron-hole transfer efficiency compared to other samples—typically signifying enhanced photocatalytic activity. The gradual increase in photocurrent intensity during the on-light intervals for the CSNPs@CdS, CSNPs, and CNPs samples may involve multiple factors: First, the slow rise of photocurrent in semiconductors (known as the relaxation phenomenon) reveals the semiconductor's sensitivity to light intensity. The Au/CdS heterojunction formed by morphological differences alters carrier flow[33]. Second, it may be influenced by the synergistic effects between carrier flow direction and effective defects. Additionally, the decrease in photocurrent intensity in CSNPs@CdS@Au can be attributed to hole accumulation. To further validate the superiority of CSNPs@CdS@Au over other structured photocatalysts in terms of carrier mobility, electrochemical impedance spectroscopy (EIS) measurements were conducted (Fig. 8f). Compared to other samples, CSNPs@CdS@Au exhibited a significantly smaller circular measurement radius. Furthermore, fitting the EIS curves yielded charge transfer resistance values of 8.59 Ω, 9.182 Ω, 12.76 Ω, and 13.33 Ω for CSNPs@CdS@Au, CSNPs@CdS, CSNPs, and CNPs, respectively. The results indicate that CSNPs@CdS@Au exhibits more efficient surface charge transfer capability, thereby achieving optimal photocatalytic activity, further confirming the superiority of the CSNPs@CdS@Au morphology.”

Fig. 8. Photocatalytic degradation of RhB by CSNPs@CdS@Au: trapping of active species, (a) p-BQ; (b) TEOA; (c) IPA; (d) Effects of scavengers on RhB degradation by CSNPs@CdS@Au; (e) transient photocurrent response curve, and (f) electrochemical impedance spectroscopy (EIS) of CSNPs@CdS@Au, CSNPs@CdS, CSNPs and CNPs.

Comment 1-5: The authors propose that charge transfer occurs from gold nanoparticles (Au NPs) to cadmium sulfide (CdS), but this claim lacks experimental validation. To substantiate this mechanism, include transient photocurrent response or electrochemical impedance spectroscopy (EIS) data to demonstrate charge transfer dynamics between Au NPs and CdS.

Response: We thank the reviewer for the kind advice. We apologize for omitting “supporting information” in our previous submission. The complete capture experiment data is now included in the main text of this submission. This figure clearly demonstrates the impact of different capture agents (TEOA, p-BQ, IPA) on degradation efficiency, providing direct experimental evidence for our conclusion that ·O₂⁻ is the primary active species and photogenerated holes (h⁺) are the secondary active species. We have added in Section 3.6 of the manuscript.

3.6. Photocatalytic mechanism

The active species of CSNPs@CdS@Au during the photocatalytic reaction were measurement via trapping experiments, various scavengers, including triethanolamine (TEOA), 1,4-Benzoquinone (p-BQ), and isopropyl alcohol (IPA) were introduced into the solution of RhB[34, 53]. The h+, ⋅O2- and ⋅OH were trapped by TEOA, p-BQ, and IPA, respectively[54]. As shown in Fig. 8a-d, the addition of p-BQ, TEOA, and IPA all resulted in a significant decrease in the catalytic rate. After 15 minutes of reaction, the degradation rates were 46.52% for p-BQ, 20.86% for TEOA, and 89.68% for IPA. The change in degradation rate was particularly pronounced when TEOA was introduced into the reaction system. It can therefore be inferred that the ⋅O2- serves as the dominant active species responsible for RhB degradation over this catalyst, while the e⁻ acts as a secondary contributor. It is noted that RhB dye itself could act as a photosensitizer in the initial stage of the photocatalytic process. However, the active species trapping experiments and the significant performance difference between catalysts (e.g., bare CdS vs. CSNPs@CdS@Au) strongly suggest that the degradation is predominantly driven by the catalysts, where the UC process and the Au/CdS Schottky junction play decisive roles in generating active radicals (mainly ·O2-).”

Fig. 8. Photocatalytic degradation of RhB by CSNPs@CdS@Au: trapping of active species, (a) p-BQ; (b) TEOA; (c) IPA; (d) Effects of scavengers on RhB degradation by CSNPs@CdS@Au; (e) transient photocurrent response curve, and (f) electrochemical impedance spectroscopy (EIS) of CSNPs@CdS@Au, CSNPs@CdS, CSNPs and CNPs.

Reviewer 2 Report

Comments and Suggestions for Authors

The manuscript, "Unique Four-Layer Core-Shell NaYF4:Yb³⁺,Er³⁺@NaYF4@CdS@Au Nanocomposites for Enhanced Full-Spectrum Photocatalytic Degradation of Rhodamine B," describes design and preparation of a unique four-layer core-shell structure catalysts and their application in photodegaradation of widely used dye, RhB. The objective is actual and interesting; however, a lot of scientific reports have been published on this subject (photodegradation of dyes).

General comments:

The experimental part is well presented and the materials obtained in each preparation step of four-layer core-shell photocatalysts were isolated. The experimental part is and easy to reproduce if needed.

Could the authors change the “peaks” to “bands” in the description of XPS and PL.

The description of the photocatalytic results should be refined to improve clarity and facilitate better understanding of the text. (Some details pointe below in the specific comments section). Could the authors avoid the not specific terms like “small amount”, “pretty small” etc., If it is possible, please give the values.

All the abbreviations used in the manuscript should be explained, for example Au100, Au10, Au50.

Specific comments presented in the order they appear in the manuscript:

“During light irradiation, aliquots of approximately 2.5 mL were drawn at quantitative time intervals, filtered to remove the photocatalyst particles, and then the concentration of RhB remaining in the supernatant was detected by UV-visible spectrophotometer (TU1810).” Could the authors provide more detailed information about the filters used, particularly regarding the filter material and pore size?

XPS characterization: “while Er and Yb are not detected as the amount of these elements is pretty small”, What about the high-resolution XPS spectra in these regions? What do the authors mean by “pretty small”? Could the authors provide the Er and Yb loading data obtained from other characterization methods? Some literature citations are missing, particularly when the authors interpret the positions of the high-resolution XPS spectra. Were the XPS spectra calibrated? If so, how?

“Fig. 5a shows the UV absorption spectra during the degradation of RhB.” Fig. 5 actually presents the UV–Vis absorption spectrum of RhB and its changes over time under light irradiation in the presence of the catalyst. Could the authors please clarify this sentence?

“As shown in Fig. 5b, the catalytic experiments of different catalysts are compared, and it can be seen that the purple line represents the catalytic rate of bare CdS.” Could the authors please clarify this description? Fig. 5b compares the kinetic curves of RhB degradation in the presence of different catalysts, rather than the experimental conditions themselves

“The black line represents CSNPs. The degradation efficiency of RhB within 15 min is approximately 5.0%, indicating that CSNPs only play a role in light conversion and do not have catalytic function.” Could the authors explain what they mean?

The control experiment, without catalysts is missing.

The comparison of the catalytic activity of the CSNPs@CdS@Au with literature dada is missing.

The stability studies of the CSNPs@CdS@Au catalyst were not performed.

It is unclear why the authors did not consider RhB as a potential photosensitizer in their mechanistic investigation of its degradation pathway.

In summary, due to the issues mentioned above, I cannot recommend this manuscript for publication in its current form.

Reviewer 3 Report

Comments and Suggestions for Authors

The manuscript entitled “Unique Four-Layer Core-Shell NaYF4:Yb³⁺,Er³⁺@NaYF4@CdS@Au Nanocomposites for Enhanced Full-Spectrum Photocatalytic Degradation of Rhodamine B” by Y. Tang and et. al. presents some interesting results on the preparation of four-layer core-shell structure and its photocatalytic properties for RhB degradation. The manuscript is well written and the conclusions. However, there are couple of issues that should be addressed prior to publication in Molecules:

  1. During the photocatalytic experiments author used 300 W xenon lamp with >420 nm cut-off filter, but there is no UV-Vis spectroscopy followed by the band gap determination to proof that the material can be activated under visible light.
  2. The author presented the luminescence properties of the NaYF4:Yb, Er. Although the results can be interesting they seem to be detached from the focus of the manuscript which is photocatalysis. I suggest either make a connection between the PL properties and the photocatalytic activity or removing the PL part entirely.
  3. The discussion on the photocatalytic properties lack any mechanistic discussion and it should be extended.

      4. The authors cite some figures that are not present in the manuscript such as figure S1-4           (probably part of the SI which is missing) .

Author Response

Dear Reviewer:

Thank you for reviewing our manuscript “Unique Four-Layer Core-Shell NaYF4:Yb³⁺,Er³⁺@NaYF4@CdS@Au Nanocomposites for Enhanced Full-Spectrum Photocatalytic Degradation of Rhodamine B” (Ms. Ref. No.: molecules-3891318) and providing constructive suggestion which have strengthened our manuscript. We have now carefully revised the manuscript according to the comment.

Review 3:

Comment 3-1: During the photocatalytic experiments author used 300 W xenon lamp with >420 nm cut-off filter, but there is no UV-Vis spectroscopy followed by the band gap determination to proof that the material can be activated under visible light.

Response: We thank the reviewer for the kind advice. We fully understand and appreciate this important point raised by the reviewers. The most compelling and direct evidence in this work comes from the photocatalytic experiments themselves: under strict conditions using a >420 nm cutoff filter, our catalytic CSNPs@CdS@Au exhibited exceptionally high RhB degradation efficiency (97.7% within 15 mins, Figure 7b). This result indisputably demonstrates the high photocatalytic activity of this catalyst system under pure visible light irradiation. The origin of its activity can be reasonably explained as follows: 1. Direct absorption of transmitted visible light (420–520 nm) by the outer CdS layer (a recognized visible-light catalyst with a bandgap ~2.4 eV). 2. The upconversion core (CSNPs) may absorb trace NIR components in the light source and convert them into UV/visible light absorbable by CdS (evidence shown in Figure 6a of the manuscript), thereby contributing to part of the activity. Although absorption spectra are lacking, the aforementioned catalytic performance was obtained under the most relevant experimental conditions (i.e., identical illumination conditions to actual catalytic testing), directly validating the material's efficacy in the target application scenario (visible-light catalysis).

Fig. 7. (a) UV-vis spectra depicting the photodegradation process of RhB solution by the CSNPs@CdS@Au composite material; (b) Degradation of RhB solution under simulated visible light; (c) Time-dependent pseudo-first-order kinetic curves; (d) Column graph of k; (e) Effect of Au content variation on CSNPs@CdS@Au; (f) Cycling performance of the CSNPs@CdS@Au photocatalyst for RhB degradation

Comment 3-2: The author presented the luminescence properties of the NaYF:Yb, Er Although the results can be interesting they seem to be detached from the focus of the manuscript which is photocatalysis. I suggest either make a connection between the PL properties and the photocatalytic activity or removing the PL part entirely.

Response: We thank the reviewer for the kind advice. We fully understand and appreciate this important point raised by the reviewers. We chose to retain the PL section as it is crucial for elucidating one of the core design principles—the upconversion process. To establish a more direct connection, we have strengthened the discussion section:We explicitly stated in Section 3.4 that the quenching of PL intensity (particularly for CSNPs@CdS and CSNPs@CdS@Au) indicates that excitation energy (especially UV emission) is efficiently absorbed by the CdS shell, which is a prerequisite for generating photogenerated carriers.The schematic diagram (Fig. 9) and corresponding description in Section 3.6 clearly depict the process where upconverted luminescence is absorbed by CdS, generating electron-hole pairs. Through these modifications, we have clearly established the logical chain: “Upconversion luminescence efficiency → CdS light absorption → Charge separation → Catalytic activity.”

Fig. 9. Schematic illustration of the proposed UC-mediated photocatalytic mechanism

Comment 3-3: The discussion on the photocatalytic properties lacks any mechanistic discussion and it should be extended.

Response: We thank the reviewer for the kind advice. We fully understand and appreciate this important point raised by the reviewers. We have substantially rewritten and expanded Section 3.6. In addition to the aforementioned PL correlations, we have: 1. Integrated results from reactive species capture experiments (Fig. 8) in greater detail. 2.Provided a more in-depth discussion of how Au NPs act as electron acceptors to form Schottky junctions, thereby promoting charge-space separation. 3.Integrated the upconversion process, CdS photoexcitation, and plasmonic effects of Au (where applicable) into a more coherent and comprehensive mechanistic model.

Fig. 8. Photocatalytic degradation of RhB by CSNPs@CdS@Au: trapping of active species, (a) p-BQ; (b) TEOA; (c) IPA; (d) Effects of scavengers on RhB degradation by CSNPs@CdS@Au; (e) transient photocurrent response curve, and (f) electrochemical impedance spectroscopy (EIS) of CSNPs@CdS@Au, CSNPs@CdS, CSNPs and CNPs.

Comment 3-4: The authors cite some figures that are not present in the manuscript such as figure S1-4 (probably part of the SI which is missing).

Response: We thank the reviewer for the kind advice. We fully understand and appreciate this important point raised by the reviewers. We sincerely apologize for the confusion caused by the omission of supporting information files in our initial submission. All supporting materials, including Figures S1–S4 and Table S1, have been submitted with this revised manuscript. We have verified and confirmed all references to these figures and tables within the main text of the manuscript. We have incorporated all the supporting information into the main text.

Round 2

Reviewer 1 Report

Comments and Suggestions for Authors

The manuscript has been substantially strengthened through the revisions, and I believe it now meets the necessary standards for publication.

Author Response

Thank you very mach. We will keep working hard.

Reviewer 2 Report

Comments and Suggestions for Authors

The authors have clarified almost all of my doubts and responded to my questions. At this point, I am satisfied.

However, the sentence:

 “In order to test the performance of CSNPs@CdS@Au photocatalyzed RhB, we prepared CSNPs, CdS, CSNPs@CdS, and CSNPs/CdS (physical mixture) as the control group, RhB as a blank control group.” requires some clarification in my opinion.

Since the composite, individual components and their physical mixture should be considered as (photo)catalysts, I would not refer to them as a "control group." Rather, the comparison of photocatalytic activities in RhB degradation between the composite, its components  and the physical mixture of the components was performed, along with an additional blank experiment without any catalysts.

If the authors agree, could they please revise the cited sentence accordingly?

Here’s a suggestion for how the cited sentence could be rewritten more clearly based on my advice:

 “To evaluate the photocatalytic performance of CSNPs@CdS@Au in the degradation of RhB, we compared it with CSNPs, CdS, CSNPs@CdS, and a physical mixture of CSNPs and CdS (CSNPs/CdS). A blank experiment without any catalyst was also conducted for reference.”

Comments on the Quality of English Language

Some sentences are unclear and require further polishing for better readability and precision.

Reviewer 3 Report

Comments and Suggestions for Authors

The authors addressed all comments/questions and the manuscript can be considered for publication in its current form.

Author Response

(The authors gave the same response as above.)
